# Prevalence of SARS-CoV-2 Omicron Sublineages and Spike Protein Mutations Conferring Resistance against Monoclonal Antibodies in a Swedish Cohort during 2022–2023

**DOI:** 10.3390/microorganisms11102417

**Published:** 2023-09-27

**Authors:** Jonathan Haars, Navaneethan Palanisamy, Frans Wallin, Paula Mölling, Johan Lindh, Martin Sundqvist, Patrik Ellström, René Kaden, Johan Lennerstrand

**Affiliations:** 1Department of Medical Sciences, Section for Clinical Microbiology and Hospital Hygiene Uppsala University, Akademiska Sjukhuset Entrance 40 Floor 5, 751 85 Uppsala, Sweden; jonathan.haars@uu.se (J.H.); johan.lindh@icm.uu.se (J.L.); patrik.ellstrom@medsci.uu.se (P.E.); rene.kaden@medsci.uu.se (R.K.); 2Chester Medical School, University of Chester, Chester CH2 1BR, UK; n.palanisamy@chester.ac.uk; 3Department of Laboratory Medicine, Clinical Microbiology, Örebro University Hospital, Södra Grev Rosengatan, 701 85 Örebro, Sweden; frans.wallin@regionorebrolan.se; 4Department of Laboratory Medicine, Clinical Microbiology, Faculty of Medicine and Health, Örebro University, 701 82 Örebro, Sweden; paula.molling@oru.se (P.M.); martin.sundqvist@oru.se (M.S.); 5SciLifeLab, Clinical Genomics Uppsala, Husargatan 3, 752 37 Uppsala, Sweden

**Keywords:** SARS-CoV-2, coronavirus, monoclonal antibodies, resistance, nanopore, Sweden, whole-genome sequencing, receptor binding domain

## Abstract

Monoclonal antibodies (mAbs) are an important treatment option for COVID-19 caused by SARS-CoV-2, especially in immunosuppressed patients. However, this treatment option can become ineffective due to mutations in the SARS-CoV-2 genome, mainly in the receptor binding domain (RBD) of the spike (S) protein. In the present study, 7950 SARS-CoV-2 positive samples from the Uppsala and Örebro regions of central Sweden, collected between March 2022 and May 2023, were whole-genome sequenced using amplicon-based sequencing methods on Oxford Nanopore GridION, Illumina MiSeq, Illumina HiSeq, or MGI DNBSEQ-G400 instruments. Pango lineages were determined and all single nucleotide polymorphism (SNP) mutations that occurred in these samples were identified. We found that the dominant sublineages changed over time, and mutations conferring resistance to currently available mAbs became common. Notable ones are R346T and K444T mutations in the RBD that confer significant resistance against tixagevimab and cilgavimab mAbs. Further, mutations conferring a high-fold resistance to bebtelovimab, such as the K444T and V445P mutations, were also observed in the samples. This study highlights that resistance mutations have over time rendered currently available mAbs ineffective against SARS-CoV-2 in most patients. Therefore, there is a need for continued surveillance of resistance mutations and the development of new mAbs that target more conserved regions of the RBD.

## 1. Introduction

Severe acute respiratory syndrome coronavirus 2 (SARS-CoV-2) is a positive sense, single-stranded RNA virus of the Coronaviridae family, which causes coronavirus disease 2019 (COVID-19). SARS-CoV-2 has caused a global pandemic and was considered a public health emergency of international concern by the World Health Organization (WHO) between 30 January 2020 and 5 May 2023 [1]. The mRNA vaccines have contributed to reducing severe disease and hospitalization, but long-term immunity remains difficult to reach due to immune escape [2]. Consequently, antiviral options such as Paxlovid [3] or monoclonal antibodies (mAbs) are necessary to further reduce the COVID-19 health burden. Hence, mAbs have become valuable in the treatment of COVID-19, especially for high-risk individuals such as immunocompromised patients. The mAbs developed against SARS-CoV-2 have been divided into four major classes, Classes I–IV, based on their epitope recognition and binding mode with the receptor binding domain (RBD) of the spike (S) protein [4,5]. The RBD lies between amino acids 333 and 527 in the S protein [6]. It interacts with angiotensin-converting enzyme 2 (ACE2) of the host cell, thereby playing a vital role in viral entry into the host cell [7]. Since 09 November 2020, mAbs have been authorized (for emergency use) for treating COVID-19 patients (Table 1).

RNA viruses, in general, tend to accumulate mutations in their genome at relatively high rates. Some of these mutations, either alone or together, can provide a selective advantage to the virus. SARS-CoV-2 started to diversify even in the first months after it was first detected [22], but was initially thought to be only slowly evolving due to the proofreading function of non-structural protein 14 [23]. Throughout the pandemic, a large number of mutations have occurred globally [23,24]. Some of these mutations in the genome of SARS-CoV-2 are known to confer significant resistance towards the licensed mAbs. Tracking such mutations helps physicians predict whether certain treatments may or may not work. Previous studies have found several mutations, mainly in the RBD of the S protein, that confer significant resistance against mAbs; among them are R346T [25,26], 371F [27], K444T [25,26], V445A/P [25,26], G446S [26], L452R [28], N460K [25,26], E484A [26], F486S/V/P [25,26], and Q493R [28,29]. Table 2 lists these and additional mutations and their corresponding resistances. These mutations have appeared in separate lineages through convergent evolution [26]. Further, treatment with mAbs can lead to or accelerate the development of resistance in the patient being treated [30,31,32,33] and may also cause these mutations to spread into the larger population [31,32].

Studying the genomic differences (i.e., mutations) between different variants and sublineages of SARS-CoV-2 is also necessary for understanding why they cause different symptoms and severities. The Omicron variant was first detected in South Africa [34] and then became the world-dominant SARS-CoV-2 variant. The Delta variant, preceding Omicron, differs from the latter variant in several ways. Cell–cell fusion and infectivity of lung and gut cells were reduced in the Omicron lineages BA.1 and BA.2 compared to that of the Delta variant [35,36]. Further, the Delta and Omicron variants differ in entry pathways [36]. Hence, identifying the mutation(s) that caused these phenotypes is important for understanding how future variants will behave and what threats they may pose. There are also observed phenotypic differences between Omicron sublineages; e.g., the BA.5 sublineages infect lung tissues more efficiently than the BA.2 sublineages, and the S protein of BA.4 and BA.5 can more efficiently fuse lung cells compared to that of BA.2 [37].

There are also other important reasons for tracking the mutations and circulating lineages of SARS-CoV-2. Tracking immune escape mutations is important for understanding the immune protection status of vaccinated or previously infected individuals, and they aid in the development of appropriate vaccines/booster vaccines. Resistance to antivirals other than mAbs may also arise in individual patients and spread across the population.

During the SARS-CoV-2 pandemic, several laboratories throughout Sweden have conducted whole-genome sequencing for contact tracing and surveillance purposes. The variants of concern and S protein mutations detected in the Uppsala region have already been published by us for the year 2021 [38], but not for the period after that. In the present study, SARS-CoV-2 genomes from COVID-19-positive samples between March 2022 and May 2023 obtained from Uppsala and Örebro regions—representing a large part of central Sweden—were sequenced using next-generation sequencing methods. Using the sequence data from these samples, we describe the prevalence of Omicron variants and S protein mutations that confer resistance against mAbs during this timeframe.

## 2. Materials and Methods

### 2.1. Samples Collection

Samples positive for SARS-CoV-2 RNA, used in this study, were collected between March 2022 and May 2023 from individuals residing in the Uppsala and Örebro regions, two of Sweden’s 21 regional councils, as part of their SARS-CoV-2 surveillance and tracing programs. The study was approved by the Swedish Ethical Review Authority under the case numbers Dnr 2022-01249-01 and 2023-02272-02. Informed consent was not applicable. Samples received were either assisted nasopharyngeal-throat swabs, saliva samples (via self-collection kit), or a combined sampling of gargled water and self-collected nasal/nasopharyngeal swabs. Further, a small number of bronchoalveolar lavage, blood, and lung biopsy samples were also included.

### 2.2. Viral RNA Extraction and Sequencing

For samples from Örebro, sequencing was performed in one of three laboratories: the Department of Laboratory Medicine, Örebro University Hospital, the National Pandemic Centre (NPC), or the Public Health Agency in Sweden (PHAS).

At Örebro University Hospital, the sequences were generated using either a MiSeq (Illumina, San Diego, CA, USA) instrument with the ARTIC V3 tiled amplicon enrichment protocol (400 bp amplicon) or with a GridION (Oxford Nanopore Technologies, Oxford, UK) instrument using the Midnight protocol based on the ARTIC network (1200 bp amplicons). Both are described in detail by Koskela von Sydow et al. [39]. Consensus sequences from Illumina data were generated with either Ridom SeqSphere + version 8.3.1–8.5.1 [40], or gms-artic version v2.0 [41]. Ridom SeqSphere+ was used with the following settings: samtools mpileup (-A -d 1000000 -B -Q 0, v1.12); ivar consensus (-q 20 -t 0.7 -m 10); ivar variants (-q 20 -m 10 -t 0.1). Gms-artic was used with the default settings and “–scheme nCoV-2019”, “–schemeVersion” “V3” or “V4.1”. Consensus sequences for the 1200 bp amplicons were generated with wf-artic version v0.3.9–v0.3.24 [42] from epi2me-labs using the default settings and –scheme_”version V1200” or “Midnight-IDT/V1”.

At the NPC, sequences were generated on the MGI DNBSEQ-G400 (MGI Tech, Shenzhen, China) using an ultraplex amplicon approach and PE100 library construction [43]. In brief, raw sequencing data were filtered using FastP [44], followed by mapping towards the reference genome. Alignments were trimmed from the primer sequence by use of iVar [45], and Variant Call Files were calculated using Freebayes [46]. Consensus sequences were generated based on major frequency bases. Low coverage areas (<30) were masked to N, and deletions (as defined by variant call) were masked, and then the sequence was collapsed at the point of deletion to keep relative genome coordinates.

At the PHAS, the sequences were generated by Genome Sequencer HiSeq (Illumina, San Diego, CA, USA), sequence mode NovaSeq 6000 S4 PE150 XP. High-quality reads were aligned to the SARS-CoV-2 reference genome (isolate Wuhan-Hu-1, MN908947) using the BWA-MEM v0.7.17-r1188 algorithm and consensus sequences were generated using consensusfixer v0.4 [47] with at least 15 supporting reads at each position. Base positions that showed less than 15x coverage were filled with Ns.

For samples from Uppsala, RNA extraction, reverse transcription PCR (RT-PCR), and Nanopore sequencing were performed at the Division of Clinical Microbiology and Hospital Hygiene, Uppsala University Hospital, Sweden. RNA extraction was performed according to the manufacturer’s instructions using Chemagic™ 360 (PerkinElmer, Waltham, MA, USA) or eMAG^®^ (bioMérieux, Marcy-l’Étoile, France) instruments. Samples positive for SARS-CoV-2 were detected with an in-house RT-qPCR method or with the BIOFIRE^®^ Respiratory 2.1 plus Panel (bioMérieux, Marcy-l’Étoile, France). After extraction, RNA eluates were stored at −20 °C. The Ct value for each sample, needed for correct dilution of the RNA, was acquired from the in-house RT-qPCR method.

Between the start of the study period and 27 June 2022, the Midnight protocol [48] was used alongside the NEBNext^®^ ARTIC SARS-CoV-2 Companion Kit protocol version 1.0_1/21 (New England BioLabs, Ipswich, MA, USA) (with a few modifications) for the library preparation and sequencing. After 27 June 2022, only the ARTIC protocol was used. The modifications to the protocol included 33 PCR cycles and the replacement of the PCR bead cleanup by a 1:10 dilution of all PCR products as per the ARTIC nCoV-2019 v3 (LoCost) protocol [49,50]. Library preparation was performed with the NEBNext^®^ ARTIC SARS-CoV-2 Companion, Native Barcoding Expansion 96 (Catalogue number: EXP-NBD196; Oxford Nanopore Technologies, Oxford, UK) and Ligation Sequencing (Catalogue number: SQK-LSK109; Oxford Nanopore Technologies, Oxford, UK) Kits. Between the start of the study period and 27 June 2022, ARTIC Network SARS-CoV-2 V3 primers were used. After this, VarSkip Short v2 primers (New England BioLabs, Ipswich, MA, USA) and BA.2 Spike-in primers [51] (New England BioLabs, Ipswich, MA, USA) were used. This switch was carried out to increase the quality of sequences since issues with the older primer sets had previously been described [38,52]. The DNA concentration of the library was measured using the Qubit HS dsDNA assay kit (Thermo Fisher, Waltham, MA, USA).

Sequencing was performed with the R9.4.1 flow cell (catalogue number: FLO-MIN106D) on a GridION instrument (Oxford Nanopore Technologies, Oxford, UK). To reduce the risk of cross-contamination between runs, flow cells were never reused. The MinKNOW software version 21.11.7–23.04.5 (Oxford Nanopore Technologies, Oxford, UK) was used with the following settings: high-accuracy base calling, barcodes on both ends, minimum barcoding alignment score = 60, minimum mid-read barcoding alignment score = 50, and trim barcodes.

Analysis of sequence data was performed with Geneious Prime version 2021.1.1 [53]. Primer sequences were trimmed using the Geneious prime BBDuk plugin version 38.84 [54] with the following settings: trim = left end, kmer length = 21, maximum substitutions = 3, trim low quality (<10) from both ends, discard reads shorter than 50 bp and custom BBDuk options; rcomp = f and restrictleft = 32. Sequence alignment and mapping to the SARS-CoV-2 Wuhan-Hu-1 reference sequence (NCBI accession number: NC_045512.2) was performed using the Geneious prime minimap2 version 2.17 [55] plugin using the following settings: data type = “Oxford Nanopore (more sensitive)”, include secondary alignments enabled, maximum secondary alignments per read = 5, minimum secondary to primary alignment score ratio = 0.8 and “remove existing trim regions from sequences” enabled. Consensus sequences were generated using the “Generate Consensus Sequence” function in Geneious Prime using the following settings: minimum coverage = 4 reads, minimum nucleotide frequency = 0.5, and “Trim to reference sequence” enabled.

The sequencing and analysis methods used here were validated at the respective laboratories and the acquired data were primarily used for national surveillance by the PHAS. The sequencing data of this study meet the quality requirements that are defined by the European Centre for Disease Prevention and Control [56]. Thus, the data, even if derived from different platforms, can be analyzed and compared in one approach.

Consensus sequences from both Örebro and Uppsala were uploaded in FASTA format to the Global Initiative on Sharing Avian Influenza Data (GISAID) database [57].

The sequences were assigned Pango lineages according to the Pango dynamic lineage nomenclature scheme [58] using the Geneious wrapper plugin for Pangolin [59] that runs the Phylogenetic Assignment of Named Global Outbreak Lineages (Pangolin) tool [60]. Unaliased Pango lineages used for grouping lineages were acquired using modified versions of R scripts contained in the PangoLineageTranslator tool [61].

To identify the mutations in the sequences, Coronapp [62] was used to find all mutations across the entire genome. Coronapp is annotation-based, which we have found necessary to find all mutations in our sequences generated from Nanopore sequencing (this approach occasionally has frameshifts).

## 3. Results

Between 1 March 2022, to 18 May 2023, 7950 SARS-CoV-2 positive samples were successfully whole-genome sequenced and analyzed, which is presented in this study. Of these samples, the majority (i.e., 6710) were obtained during 2022 and the remaining (i.e., 1240) were from 2023. Approximately half of these samples (i.e., 4478) were from the Uppsala region and the rest (i.e., 3472) were from the Örebro region. The distribution of samples sequenced over time is shown in Figure 1.

The Pangolin tool was used to determine the Pango lineage of the samples. The week-wise abundance (in %) of particular sublineages was calculated relative to the total number of SARS-CoV-2 positive samples sequenced that week. Figure 2 shows the dynamics of different SARS-CoV-2 sublineages in central Sweden during the study period. The Omicron BA.2 sublineage and its descendants dominated from the start of our study in March 2022 until the middle of May 2022 when it was replaced, mainly by the BA.5 sublineage and its descendants, but to some degree also by the BA.4 sublineage. The BA.5 and its descendants, including BE, BF, and BQ sublineages, were the main sublineages between June 2022 and December 2022. After this, the situation was more dynamic, with a mix of BA.2 and BA.5 lineages. The most prominent of the sublineages that increased during this time were the BA.2.75 (including BN and CH sublineages) and BQ sublineages. The XBB sublineages began to take over and became the most common ones by the late spring of 2023. The XBB sublineage is the product of a recombination event between the two sublineages BJ.1 and BM.1.1.1, which are both descendants of BA.2. [63]. The increase in BQ and XBB sublineages is important because of their known resistance to the available mAbs [25]. The number of sequences assigned to the XBB.1.9.1 lineage increased towards the end of the study period (Figure 2) and was the most common Pango lineage in May 2023 (Appendix A), the last month of the study period.

Among the amino acid (aa) mutations in the RBD that are involved in mAb resistance, several changed in their relative abundance throughout the study period (Figure 3). The over-time relative abundance of all aa substitutions in the S protein of SARS-CoV-2 from samples collected during the period March 2022 to May 2023 can be seen in Appendix A.

Mutations in the SARS-CoV-2 S protein, especially at positions 346 and 444, mainly R346T and K444T, increased during the autumn of 2022 (Figure 4a), indicating an increase in infections that were resistant to mAb treatment with tixagevimab and cilgavimab. There was a simultaneous increase in the number of samples with double mutations at aa positions 346 and 444, and samples with single mutations at one of the two positions, showing that several different lineages increased in frequency at the same time but were conferring similar mAb resistance through different mutations. Mutations in position 444, mainly K444T, increased during the autumn of 2022 but decreased by the spring of 2023, and mutations in position 445, mainly V445P, increased during the autumn of 2022 and spring of 2023 (Figure 4b). K444T and V445P are involved in resistance to bebtelovimab.

## 4. Discussion

Since the start of the COVID-19 pandemic, SARS-CoV-2 has evolved rapidly and still continues to evolve. The changes that have occurred during this time have affected many aspects of the virus and the COVID-19 disease outcome. Since virus transmission and COVID-19 symptoms/severities vary between different sublineages [35,36,37], it is important to have a good knowledge of sublineage(s) that are in circulation locally at a particular time. This will benefit in treating patients effectively and also in implementing effective control measures to prevent further spread. Using SARS-CoV-2 sequences collected between March 2022 and May 2023 from individuals residing in the Uppsala and Örebro regions of central Sweden, the dynamics of various sublineages in the regions and aa mutations that are relevant regarding mAbs are reported in this study. 

We found that the mutations that occurred during the study period, especially in the RBD, have a significant effect on the usefulness of available mAbs [25,26].

Treatment with bamlanvimab, regdanvimab, casirivimab/imdevimab, or sotrovimab has been unlikely to be successful in the Uppsala and Örebro regions throughout the entire study period. Mutations such as L452R, E484A, E484K, F486V, or Q493R, either alone or in combination, confer a high-fold resistance (>400 fold-change range) against bamlanvimab (Table 2) and these mutations have been present in most samples during the study period (Figure 3). Some of the aforementioned mutations, L452R and Q493R, also confer high-fold resistance to regdanvimab. While Q493R, which confers the highest resistance to regdanvimab of all the aa mutations (as a single mutation 949-fold) in the RBD of the S protein, became rare after the first half of 2022, other mutations such as S371F and E484A are very common and L452R was very common between June 2022 and March 2023 (Figure 3). The common E484A mutation also confers medium-fold resistance (7.8-fold) to the combination of casirivimab and imdevimab (Table 2). In addition, the mutations K444T, V445P, and G446S, which became common during the end of 2022 and the start of 2023 (Figure 3), also confer low- to medium-fold resistance to this mAb combination. The S371F mutation, on the other hand, confers medium fold-change resistance (i.e., 9.7-fold) against sotrovimab. This left tixagevimab and cilgavimab as the only available option in the EU while the USA also had access to bebtelovimab.

During the second half of 2022, the number of samples with mutations in 346 or 444 increased rapidly, from being present in <10% of samples to being present in >90% of samples in just four months (Figure 4a). This increase was mainly due to the aa mutations R346T and K444T which were much more common than R346S, R346K, R346I, K444R, or K444N (Figure 3). The simultaneous increase of mutations in S protein aa 346, 444, or both shows that the prevalence of mAb resistance increased by the spread of several different lineages carrying different sets of mutations with similar phenotypic effects. This taken together with the fact that both sublineages originating from BA.2 and sublineages originating from BA.5 spread at the same time and showed similar mutations indicates that parallel evolution was and had been taking place. The proportion of samples with either a mutation in 444 or 445 increased rapidly during the last months of 2022 and the beginning of 2023 (Figure 4b). The increase in aa mutations at position 445 was mainly because of an increase of V445P (Figure 3) which is present in all XBB lineages. Mutations in aa positions 346 and/or 444 have been shown to cause high-fold resistance (>200) against the mAb treatment with tixagevimab and cilgavimab [25] used in Evusheld which was the main mAb treatment available in Europe during this time. In the USA, bebtelovimab was available and if it had been available in Uppsala and Örebro, it could have been used to treat patients until the mutations K444T, and/or V445P, which cause high-fold resistance (>6000) to bebtelovimab [25,26] also became very common (Figure 4b). However, targeted testing and whole-genome sequencing of SARS-CoV-2 samples from candidates for bebtelovimab treatment could have been performed for several months before the 445 aa changes became fixed in the SARS-CoV-2 population. This would have allowed more patients to be treated with mAbs.

The abundance of sublineages BN.1, BQ.1, CH.1, XBB, and their descendants which are known for their high resistance against the mAbs that were available during 2022 and the first half of 2023 [25] increased among our samples during the end of 2022 and had become very dominant by the start of 2023 (Figure 2). During 2023, sublineages with high resistance to mAbs remained dominant but the composition of sublineages changed. Descendants of XBB have taken over and very few samples sequenced from April to May 2023 are descendants of BA.5. By the final week of this study, XBB.1.9.1 had become the most common sublineage (Appendix A). This increase of XBB.1.9.1 as well as that of XBB.1.5 (Figure 2) is likely because of the mutation F486P which increased the ACE2 binding affinity and is a strong neutralizing antibody evading mutation [64]. This suggests that future studies should focus on investigating the XBB sublineages and how mutations with an XBB genetic background affect any new vaccines, mAbs or other antivirals. However, parallel evolution can take place in different distantly related sublineages and replacements with completely new sublineages have happened rapidly and repeatedly, so investigation of other sublineages is also well justified. Treatment with currently available mAbs is unlikely to be successful for most current SARS-CoV-2 patients, but some patients with chronic infections may still carry SARS-CoV-2 variants who are susceptible to available mAbs. If samples from these patients are whole-genome sequenced, it could help determine which mAb would be effective for their treatment.

While currently available mAbs may be ineffective for treating most SARS-CoV-2 patients, new mAbs are being developed, such as AZD3152 by AstraZeneca which is in clinical trials with an estimated primary completion date of 7 September 2023 [65]. This antibody targets a conserved region of the RBD [66] and is active against currently circulating sublineages that are resistant to other mAbs (Table 2).

The repeated events of evolved resistance to mAbs means that the development of new mAbs and other antiviral treatments remains important for patients infected with SARS-CoV-2. The rapid increase in resistance to tixagevimab and cilgavimab highlights that these changes need to be communicated quickly by labs and scientists to the public. Therefore, the continued surveillance of SARS-CoV-2 through whole-genome sequencing is essential for understanding the evolution of the virus and for providing scientists, physicians, patients, decision makers, and drug manufacturers with correct and updated information for curbing this infection in the population.

## Figures and Tables

**Figure 1 microorganisms-11-02417-f001:**
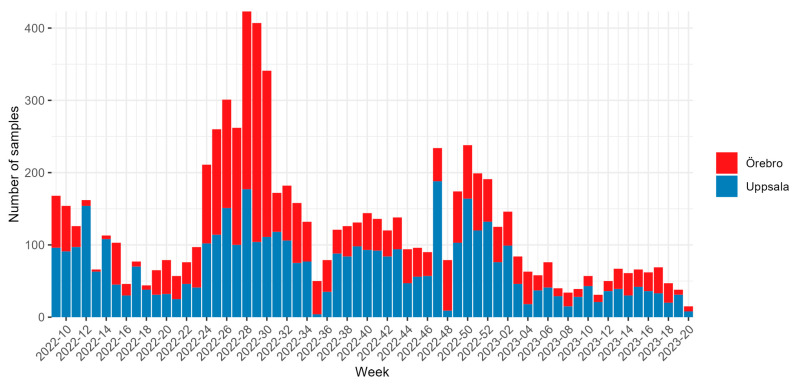
Week-wise distribution of the number of SARS-CoV-2 positive samples from Örebro and Uppsala regions that were sequenced by us. x-axis: year-week.

**Figure 2 microorganisms-11-02417-f002:**
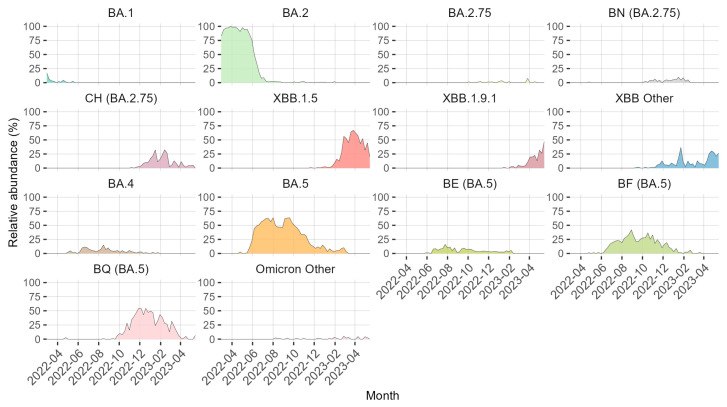
Relative abundance (in %) of different SARS-CoV-2 sublineages over time in Uppsala and Örebro regions during the period March 2022 to May 2023 with weekly time points. Note: Multiple Pango lineages have been grouped together into groups based on ancestry. x-axis: year-month.

**Figure 3 microorganisms-11-02417-f003:**
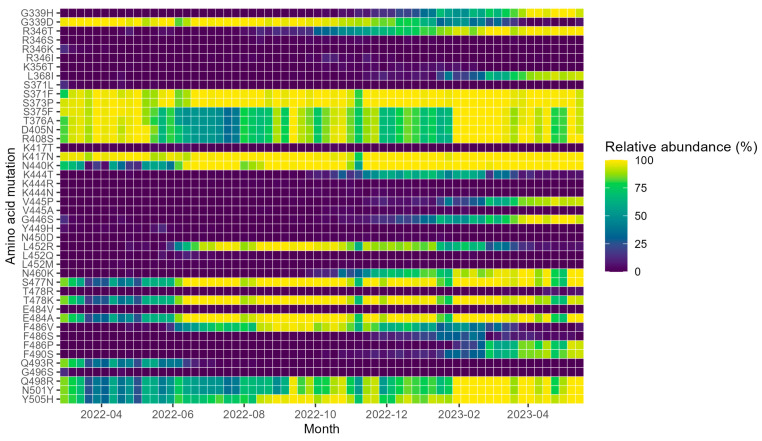
Heatmap showing the relative abundance (in %) of amino acid mutations in the receptor binding domain (RBD) of the SARS-CoV-2 S protein for each week between March 2022 to May 2023. Amino acid mutations that were present in less than 10 samples have been excluded. x-axis: year-month.

**Figure 4 microorganisms-11-02417-f004:**
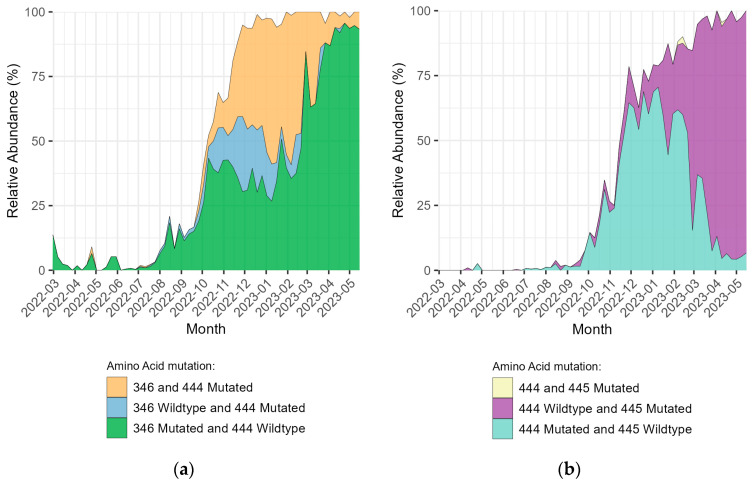
Relative abundance per week (in %) from March 2022 to May 2023 of SARS-CoV-2 sequences containing amino acid (aa) mutations in positions 346, 444, and 445 of the Spike (S) protein. Mutations in other positions are present in these sequences but not shown here. (**a**) Positions 346 and/or 444 (mainly R346T and K444T), are relevant for resistance against tixagevimab and cilgavimab. (**b**) Positions 444 and/or 445 (mainly K444T and V445P), are relevant for resistance against bebtelovimab. x-axis: year-month.

**Table 1 microorganisms-11-02417-t001:** Monoclonal antibodies (mAbs) for the treatment of COVID-19 that have been granted emergency use authorization (EUA) by the American Food and Drug Administration (FDA) or under European Medicines Agency (EMA) rolling review.

Monoclonal Antibody	Commercial Name	Class	Granted Emergency Use by the FDA	Revised Emergency Use by the FDA	EMA Rolling Review Started	EMA Rolling Review Stopped
bamlanvimab	NA	II	9 November 2020 [8]	16 April 2021 [8]	11 March 2021 [9]	29 October 2021 [10]
bamlanvimab and etesevimab	NA	II and I	9 February 2021 [11]	NA	11 March 2021 [9]	29 October 2021 [10]
regdanvimab	Regikrona	I	NA	NA	12 November 2021 [12]	NA
casirivimab and imdevimab	Ronapreve/REGEN-COV	I and III	21 November 2020 [13]	24 January 2022 [13]	NA	NA
sotrovimab	Xevudy	III	26 May 2021 [14]	5 April 2022 [15]	17 December 2021 [16]	NA
tixagevimab and cilgavimab	Evusheld	I and II	8 December 2021 [17]	26 January 2023 [18]	25 March 2022 [19]	NA
bebtelovimab	NA	III	11 February 2022 [20]	30 November 2022 [21]	NA	NA

NA = not applicable.

**Table 2 microorganisms-11-02417-t002:** SARS-CoV-2 receptor binding domain (RBD) mutations and their fold-resistance towards currently available mAbs *.

Omicron Lineage or Mutants with a Single RBD Mutation	** Important Resistance Mutations in Lineage	Bamlanivimab	Regdanvimab	Casirivimab and Imdevimab	Sotrovimab	Tixagevimab and Cilgavimab	Bebtelovimab	AZD3152
BA 2	S371F + T478K + E484A + Q493R	>1000	>1000	387	21	8	1	0.6
BA 2.75	S371F + G446S + N460K + E484A	>1000	42	>1000	12	24	3.1	1.9
BA 4	S371F + L452R + E484A + F486V	>1000	>1000	25	22	25	1	0.2
BA 5	S371F + L452R + E484A + F486V	>1000	>1000	25	22	25	1	0.2
BE (BA.5)	S371F + L452R + E484A + F486V	NI	NI	NI	NI	NI	NI	NI
BN (BA 2.75)	R346T + S371F + G446S + N460K + E484A	NI	NI	NI	NI	NI	NI	NI
BQ (BA.5)	S371F + K444T + L452R + N460K + E484A + F486V	NI	NI	200	26	>1000	900	0.9
CH (BA 2.75)	R346T + S371F + K444T + G446S + E484A	NI	NI	>1000	16	>1000	>1000	NI
XBB	R346T + S371F + V445P + G446S + N460K + E484A + F486S	>1000	NI	200	14	738	>1000	0.3
XBB 1.5	R346T + S371F + V445P + G446S + N460K + E484A + F486P	NI	NI	751	18	867	475	0.2
XBB 1.9.1	R346T + S371F + V445P + G446S + N460K + E484A + F486P	NI	NI	NI	NI	NI	NI	NI
R346T	R346T	NI	NI	NI	1.3	NI	NI	NI
S371F	S371F	0.9	21	0.6	9.7	0.6	2.2	NI
K444T	K444T	NI	NI	6.2	NI	NI	>1000	NI
V445A	V445A	NI	NI	4.7	3.4	NI	83	NI
G446S	G446S	1.2	1.1	4.5	1.6	2	2	NI
L452R	L452R	>1000	35	2	1.1	1	0.6	NI
N460K	N460K	1.5	NI	1.3	1.2	2	1.2	NI
E484A	E484A	697	7.9	7.8	0.9	5.1	1.4	NI
E484K	E484K	>1000	1.4	1.5	0.4	3.6	0.7	NI
F486V	F486V	490	NI	2.7	1.1	9.5	1.5	NI
Q493R	Q493R	>1000	949	NI	1.3	3.4	1	NI

* Data were taken from Stanford University’s coronavirus antiviral and resistance database [34,35]. NI = no information. ** Important mutations in the RBD that confer significant resistance against mAbs are R346T, S371F, D405N, K417N, K444T, V445A/P, G446S, L452R, N460K, T478K, E484A, F486S/V/P, and Q493R.

## Data Availability

Consensus sequences for all samples are available on GISAID as EPI_SET_230905gu, doi:10.55876/gis8.230905gu.

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
