# Peer review of "Prevalence of SARS-CoV-2 Omicron Sublineages and Spike Protein Mutations Conferring Resistance against Monoclonal Antibodies in a Swedish Cohort during 2022–2023"

_microorganisms, 2023, doi:10.3390/microorganisms11102417_

Round 1

Reviewer 1 Report

Summary:

This study by Haars et al. describes the prevalence of Omicron variants in two regions of Sweden over the course of about a year. The authors focus on variants that were known for providing SARS-CoV-2 with resistance against certain monoclonal antibodies. 7,950 positive samples were included in the study and variants of interest were selected based on published literature.

Comments:

-       The title suggests that data regarding resistance to monoclonal antibodies was generated as part of this study, however, data regarding monoclonal antibodies was derived from the results of published studies. While the focus here is clearly on monoclonal antibodies, I would suggest a title change reflecting the fact that the authors focused on known mutations.

-       I understand that all the samples processed in Uppsala were sequenced using Nanopore sequencing however, the methods varied quite a lot across sites, so I would recommend either listing all the technologies or mentioning a broader term encompassing all the technologies used, at least in the abstract.

-       All sequencing methods come with their unique sets of strengths and weaknesses. Could the authors comment on potential sensitivity and specificity-related concerns, how those might differ across sites, and if this could introduce a bias?

-       Along the same lines, in Figure 1, the breakdown per site could be shown, using stacked colored bars, for example, to illustrate the fact that samples collected across the sites were not collected over non-overlapping/different time periods.

Author Response

We appreciate the comments which we find helpful and constructive. As a first response we would like to point out that we have made some further minor changes throughout the manuscript to make the study clearer.

Comments:

-       The title suggests that data regarding resistance to monoclonal antibodies was generated as part of this study, however, data regarding monoclonal antibodies was derived from the results of published studies. While the focus here is clearly on monoclonal antibodies, I would suggest a title change reflecting the fact that the authors focused on known mutations.

Response: This has been revised with a new title “Prevalence of SARS-CoV-2 Omicron Sublineages and Spike Protein Mutations Conferring Resistance Against Monoclonal Antibodies in a Swedish Cohort During 2022-23”

-       I understand that all the samples processed in Uppsala were sequenced using Nanopore sequencing however, the methods varied quite a lot across sites, so I would recommend either listing all the technologies or mentioning a broader term encompassing all the technologies used, at least in the abstract.

Response: This has been adjusted in the abstract.

-       All sequencing methods come with their unique sets of strengths and weaknesses. Could the authors comment on potential sensitivity and specificity-related concerns, how those might differ across sites, and if this could introduce a bias?

Response: This has been revised as suggested in the Materials and Methods part, fourth last paragraph.

-       Along the same lines, in Figure 1, the breakdown per site could be shown, using stacked colored bars, for example, to illustrate the fact that samples collected across the sites were not collected over non-overlapping/different time periods.

Response: We have adjusted Figure 1 as suggested.

Reviewer 2 Report

I really enjoyed this manuscript. It highlights the persistent mutations in some of the amino acids in RBD of the S protein. Showing the evolutionary journey of the SARS-CoV-2 virus in the Swedish cohort over the past year. This provides valuable insights for current and future monoclonal antibodies R&D.

My only comment, or suggestion, is, it would be better if the authors could access more clinical data from other regions of the world, it might provide a deeper understanding of the virus's evolutionary trajectory. Additionally, analyzing sequences of the other proteins besides the S protein could also enhance our comprehension of the virus's evolution and help predict its potential future behavior changes.

Of course, this is already a very well-organized manuscript, and I believe it meets the standards of this Journal for direct publication.

Author Response

We appreciate the kind words and comment which we find helpful and constructive. As a first response we would like to point out that we have made some changes in the Title, Table 1, Table 2, Figure 1, Figure 4 and minor changes through the manuscript to make the study clearer.

My only comment, or suggestion, is, it would be better if the authors could access more clinical data from other regions of the world, it might provide a deeper understanding of the virus's evolutionary trajectory. Additionally, analyzing sequences of the other proteins besides the S protein could also enhance our comprehension of the virus's evolution and help predict its potential future behavior changes.

Response: Thank you for this suggestion but looking at more clinical data from other regions and analyzing mutations outside the S-protein will be a whole new study that will take too much time. However, we will have such study in mind to be published separately.